# Effects of outdoor play on body composition and physical performance in children: the Yamanashi Adjunct study of the Japan Environment and Children's Study

**Masanori Wako** [1]*, **Taro Fujimaki**[1], **Jiro Ichikawa**[1], **Kensuke Koyama**[2], **Ryoji Shinohara**[3], **Sanae Otawa**[3], **Anna Kobayashi**[3], **Sayaka Horiuchi**[3], **Megumi Kushima**[3], **Zentaro Yamagata** [3], **Hirotaka Haro**[1], **on behalf of the Yamanashi Adjunct Study of the Japan Environment and Children's Study Group**[¶]

1 Department of Orthopaedic Surgery, Faculty of Medicine, University of Yamanashi, Chuo-shi, Yamanashi, Japan, 2 Department of Orthopaedic Surgery, Nirasaki City Hospital, Honmachi, Nirasaki-shi, Yamanashi, Japan, 3 Center for Birth Cohort Studies, Interdisciplinary Graduate School of Medicine, University of Yamanashi, Chuo-shi, Yamanashi, Japan

¶ Membership of the Yamanashi Adjunct Study of the Japan Environment and Children's Study Group is provided in the Acknowledgments.
* wako@yamanashi.ac.jp

## Abstract

### Introduction

Childhood is a pivotal developmental stage that substantially affects lifelong habits. Recent research has emphasized the vital role of outdoor play in children's mental and physical well-being. Despite the World Health Organization recommending 1 hour of daily physical activity for children, a knowledge gap exists regarding the specific link between children's physical performance, body composition (evaluated through bioelectrical impedance analysis [BIA]), and outdoor play habits.

### Methods

Utilizing data from the Japan Environment and Children's Study, a national birth cohort study, this study included 494 eight-year-old participants. The assessment included body composition (height, weight, body fat percentage, predicted muscle weight, and phase angle using BIA) and physical performance (50 m sprint, standing long jump, 20 m shuttle run, and handgrip strength). Parents provided information on children's outdoor playtime.

### Results

The group with more outdoor play demonstrated superior sports test results, particularly among boys. Girls engaged in increased outdoor play exhibited higher predicted muscle weights, whereas boys showed greater phase angles in the lower limbs. Handgrip strength correlated with phase angle and predicted muscle weight. Notably, the association between body composition and sports test results was more pronounced in boys than in girls, with phase angles exhibiting stronger links to running and jumping.

**Data availability statement:** We have uploaded the minimal data set as supplementary information (S1 Table) and included captions at the end of the manuscript.

**Funding:** The author(s) received no specific funding for this work.

**Competing interests:** The authors have declared that no competing interests exist.

## Conclusion

This pioneering study explored the relationship between outdoor play, body composition, and physical performance in children. Outdoor play positively correlated with improved sports performance, revealing sex disparities in body composition changes. Unlike previous studies focusing on general physical development, this study scrutinized specific physical functions, uncovering correlations between phase angle and muscle quality. Findings suggest that outdoor play positively impacts muscle quality, especially in boys, contributing to enhanced physical performance in children. Understanding these effects on body composition and physical activity is imperative for promoting children's health.

## Introduction

Childhood is a critical period in human development characterized by rapid growth, maturation, and the establishment of lifelong habits. Recently, research has emphasized the importance of adequate outdoor play for the mental and physical development of children [1–3]. Outdoor play gives children the opportunity to be physically active and exposes them to the natural environment, influencing their overall health. The World Health Organization recommends at least 1 hour of moderate-to-vigorous daily physical activity for children [4]. However, reports on the relationship between children's specific physical performance and outdoor play habits are rare.

The phase angle (PhA) shows the geometric relationship between resistance and reactance, constituting the raw data for bioelectrical impedance (BIA) [5]. BIA is a body composition analysis method that uses low-frequency currents to estimate body composition. The theoretical assumption is that the human body is a network of resistors (body fluids) and capacitors (cell membranes) [6]. Resistance is caused by body fluids and electrolytes, whereas reactance is the ability of membranes to maintain their potential. Several reports have recently indicated that BIA is useful for the qualitative assessment of muscles [7–10]. Although several reports have been published on various physical abilities and BIA in adults [11–13], few reports exist on children's physical performance and body composition, including BIA.

Understanding the relationship between children's outdoor play, body composition (including BIA), and detailed physical performance is crucial for comprehending which aspects of body composition are influenced by outdoor play and how they relate to motor performance. This provides a more concrete understanding of the significance of outdoor play in children's motor development.

This study aimed to investigate how children's outdoor playtime affects their body composition and physical performance, examining the correlation between body composition and physical performance.

## Materials and methods

### Study design

The Japan Environment and Children's Study (JECS) is a national project funded by the Ministry of Environment. This birth cohort study was undertaken to elucidate the influence of environmental factors during the fetal period and early childhood on children's health, with follow-up until adolescence. The details of the JECS protocol and baseline data are available elsewhere [14]. Besides this nationally conducted cohort study, an adjunct study was

conducted among the JECS participants who were 8 years old at our institution. This adjunct study included ophthalmologic or oral investigations, body composition tests, including BIA using a body composition analyzer, and postural stability tests using a foot pressure plate. The participants and their respective data were used in this study.

This study was approved by the Ethics Committee of the University of Yamanashi (approval number: 2020). Written informed consent was obtained from all participants' guardians in accordance with the Declaration of Helsinki.

### Participants

A total of 1,340 children aged 8 years participated in the adjunct study conducted at our institution between November 2020 and November 2022. Of these 1,340 participants, 846 were excluded owing to inaccurate measurements caused by physical or mental reasons like hyperactivity. Additionally, those lacking elementary school sports test results in the questionnaire were excluded. Ultimately, 494 children (260 girls and 234 boys) were included in this study.

### Body composition assessment

All body composition measurements were performed at our institution. The patient height (HT) was measured and recorded in centimeters to the nearest millimeter. Body weight (BW), body fat percentage (BF), predicted muscle weight (MW) (trunk, right upper limb [RU], left upper limb [LU], right lower limb [RL], and left lower limb [LL]), and PhA (RU, LU, RL, and LL) were assessed using the MC-780U body composition analyzer (TANITA, Tokyo, Japan). This analyzer generated analyses based on the BIA observed while the participants wore shorts and a T-shirt. Before measuring the body composition on the analyzer, the soles of the participants were thoroughly wiped with wet tissue.

### Physical performance assessment

Regarding physical performance, we asked parents to save the results of sports tests administered that year at each participant's elementary school, which were later collected and calculated using a questionnaire. In Japan, sports tests for primary school children are annually conducted nationwide according to uniform standards, and data on the 50 m splinting time (50S), standing long jump distance (SLJ), 20 m shuttle run test (20SR), and hand grip strength (HG) were used from these tests. Briefly, the 20SR was a field-based progressive aerobic exercise test that involved continuous running between two lines 20 m apart in time. The initial speed was 8.5 km/h, increasing 0.5 km/h every minute. The children were encouraged to continue running for as long as possible during the test.

### Outdoor playtime

The parents provided survey data on whether the participants played outside for at least 1 hour daily.

### Statistical analysis

Sex differences between each survey item were evaluated using t-tests. Additionally, differences in each item, based on whether participants played outside for at least 1 hour each day, were evaluated using a t-test. Correlations between body composition and sports test results were assessed using Pearson correlation coefficients. Statistical significance was set at $p < 0.05$.

## Results

Table 1 summarizes each survey item for all participants. The sex differences in PhA were significant in LU and RL, whereas not in other items. The MW was significantly greater in boys than in girls, mainly in the lower limbs. Sports test results were superior in boys except for the HG.

Table 2 shows the differences in body composition and sports test results based on whether the participants played outside for at least 1 hour each day. In girls, PhA did not differ by time spent playing outside; however, the MW of the RU, RL, and LL were significantly higher in the group that played more outside, and the MW of the trunk and LU was greater in the group, although this was not significant. Sports test results showed that the 50S and SLJ were significantly better in the group with more outdoor play. In boys, unlike girls, the lower limb PhA was greater in the group with more outdoor playtime. Moreover, MW did not differ significantly between the two groups. Sports test results were significantly better in the group with more outdoor playtime in all categories.

Table 3 shows the correlations between body composition and the sports test results. HG showed a significantly moderate correlation with MW and a weak correlation with PhA in both sexes. Other sports test results showed no correlation with MW; however, they showed a weak but significant correlation with PhA in the lower limbs, particularly in boys.

## Discussion

This is the first study to report the relationship between outdoor play, body composition, and physical performance in children.

Overall, the group with more outdoor play had better results on the sports tests. Significant differences were found for girls in the 50S and SLJ and boys in all disciplines. Regarding body composition, MW in girls and PhA in boys were greater in the group that played more outside. HG correlated with PhA and MW; however, other sports test results did not correlate

**Table 1. Examined items and their sex differences.**

|  |  | Total (n = 494) | Girls (n = 260) | Boys (n = 234) | p value |
|---|---|---|---|---|---|
| **HT (cm)** |  | 125.3 ± 4.9 | 125.0 ± 5.1 | 125.6 ± 4.6 | 0.1940 |
| **BW (kg)** |  | 25.4 ± 4.7 | 25.4 ± 5.2 | 25.5 ± 4.2 | 0.7488 |
| **BF (%)** |  | 14.3 ± 6.4 | 15.2 ± 6.4 | 13.4 ± 6.3 | 0.0013 |
| **Phase angle** | **RU** | 6.4 ± 2.4 | 6.6 ± 2.8 | 6.3 ± 2.0 | 0.1948 |
|  | **LU** | 6.0 ± 2.6 | 6.2 ± 3.1 | 5.8 ± 1.9 | 0.0444 |
|  | **RL** | 5.0 ± 0.5 | 5.0 ± 0.5 | 5.1 ± 0.5 | 0.0408 |
|  | **LL** | 5.1 ± 0.5 | 5.1 ± 0.5 | 5.1 ± 0.5 | 0.6437 |
| **MW (kg)** | **trunk** | 11.7 ± 1.4 | 11.6 ± 1.5 | 11.9 ± 1.3 | 0.0123 |
|  | **RU** | 0.8 ± 0.1 | 0.8 ± 0.2 | 0.8 ± 0.1 | 0.5738 |
|  | **LU** | 0.8 ± 0.2 | 0.8 ± 0.2 | 0.8 ± 0.1 | 0.4990 |
|  | **RL** | 3.6 ± 0.5 | 3.5 ± 0.5 | 3.7 ± 0.4 | <0.0001 |
|  | **LL** | 3.5 ± 0.5 | 3.5 ± 0.5 | 3.6 ± 0.4 | 0.0002 |
| **Sports test** | **HG (kg)** | 10.4 ± 2.5 | 10.2 ± 2.5 | 10.6 ± 2.4 | 0.1215 |
|  | **50S (seconds)** | 10.9 ± 1.1 | 11.0 ± 1.2 | 10.7 ± 1.0 | 0.0009 |
|  | **SLJ (m)** | 121.7 ± 18.9 | 118 ± 18.4 | 125.8 ± 18.7 | <0.0001 |
|  | **20SR** | 22.7 ± 11.1 | 20.4 ± 9.6 | 25.2 ± 12.0 | <0.0001 |

HT; height, BW; body weight, BF; body fat, RU; right upper limb, LU; left upper limb, RL; right lower limb, LL; left lower limb, MW; muscle weight; HG; hand grip, 50S; 50m splint time, SLJ; standing long jump, 20SR; 20m shuttle run.

**Table 2. Each study item divided by outdoor playtime.**

**A. Girls**

| | | Outdoor play < 1 h/day (n = 166) | Outdoor play ≥ 1 h/day (n = 93) | p value |
|---|---|---|---|---|
| **HT (cm)** | | 124.4 ± 4.8 | 126.0 ± 5.4 | 0.0111 |
| **BW (kg)** | | 24.9 ± 5.0 | 26.3 ± 5.5 | 0.0325 |
| **BF (%)** | | 14.8 ± 6.0 | 16.1 ± 7.0 | 0.1097 |
| **Phase angle** | RU | 6.51 ± 2.45 | 6.65 ± 3.29 | 0.6885 |
| | LU | 6.34 ± 3.15 | 6.08 ± 2.91 | 0.5085 |
| | RL | 4.97 ± 0.48 | 5.08 ± 0.50 | 0.0640 |
| | LL | 5.01 ± 0.53 | 5.12 ± 0.54 | 0.1339 |
| **MW (kg)** | trunk | 11.45 ± 1.49 | 11.79 ± 1.50 | 0.0815 |
| | RU | 0.83 ± 0.17 | 0.88 ± 0.19 | 0.0320 |
| | LU | 0.84 ± 0.19 | 0.87 ± 0.19 | 0.1935 |
| | RL | 3.40 ± 0.45 | 3.56 ± 0.49 | 0.0064 |
| | LL | 3.40 ± 0.46 | 3.55 ± 0.50 | 0.0143 |
| **Sports test** | HG (kg) | 10.0 ± 2.6 | 10.5 ± 2.3 | 0.2340 |
| | 50S (seconds) | 11.2 ± 1.3 | 10.8 ± 101 | 0.0250 |
| | SLJ (m) | 116.0 ± 17.3 | 121.6 ± 19.8 | 0.0208 |
| | 20SR | 19.6 ± 8.9 | 21.7 ± 10.1 | 0.0971 |

**B. Boys**

| | | Outdoor play < 1 h/day (n = 134) | Outdoor play ≥ 1 h/day (n = 96) | p value |
|---|---|---|---|---|
| **HT (cm)** | | 125.3 ± 4.6 | 125.9 ± 4.8 | 0.4218 |
| **BW (kg)** | | 25.4 ± 4.4 | 25.6 ± 3.9 | 0.7752 |
| **BF (%)** | | 13.1 ± 6.8 | 13.1 ± 5.5 | 0.6256 |
| **Phase angle** | RU | 6.32 ± 2.11 | 6.23 ± 1.80 | 0.7269 |
| | LU | 5.79 ± 2.00 | 5.77 ± 1.78 | 0.9436 |
| | RL | 5.02 ± 0.55 | 5.19 ± 0.50 | 0.0166 |
| | LL | 4.99 ± 0.54 | 5.18 ± 0.52 | 0.0075 |
| **MW (kg)** | trunk | 11.78 ± 1.45 | 12.05 ± 1.15 | 0.1288 |
| | RU | 0.84 ± 0.11 | 0.84 ± 0.11 | 0.9764 |
| | LU | 0.83 ± 0.12 | 0.85 ± 0.13 | 0.2417 |
| | RL | 3.65 ± 0.43 | 3.68 ± 0.42 | 0.5308 |
| | LL | 3.59 ± 0.43 | 3.63 ± 0.42 | 0.4392 |
| **Sports test** | HG (kg) | 10.1 ± 2.3 | 11.3 ± 2.5 | 0.0001 |
| | 50S (seconds) | 10.9 ± 1.1 | 10.4 ± 0.7 | 0.0002 |
| | SLJ (m) | 123.4 ± 18.8 | 128.8 ± 18.4 | 0.0327 |
| | 20SR | 22.8 ± 11.2 | 28.4 ± 12.3 | 0.0006 |

HT; height, BW; body weight, BF; body fat, MW; muscle weight, RU; right upper limb, LU; left upper limb, RL; right lower limb, LL; left lower limb, HG; hand grip, 50S; 50m splint time, SLJ; standing long jump, 20SR; 20m shuttle run.

with either PhA of the upper limb or MW. However, they did correlate with PhA of the lower limb, with this correlation being stronger in boys.

Although many reports exist on outdoor play and the health and physical development of children, few have explored the relationship between specific physical functions and outdoor play. Kwon et al. reported that the more time children spent playing outside, the better their gross motor development, especially in object control (throwing and kicking a ball); however, no differences were observed in locomotor items such as running and jumping. The current study found significant differences in running and jumping, in contrast to the findings of

**Table 3. Correlations between sports test results and body composition.**

**A. Girls**

| | | Sports test | | | |
|---|---|---|---|---|---|
| | | HG | 50S | SLJ | 20SR |
| Phase angle | RU | 0.1925** | -0.0436 | 0.0897 | 0.0687 |
| | LU | 0.1323** | -0.0321 | 0.0392 | 0.1125 |
| | RL | 0.2564** | -0.2280** | 0.1383* | 0.1418* |
| | LL | 0.2391** | -0.1846** | 0.1084 | 0.153* |
| MW | trunk | 0.3775** | 0.0909 | -0.0833 | -0.1069 |
| | RU | 0.4383** | 0.0096 | 0.0438 | -0.0675 |
| | LU | 0.4101** | 0.0344 | 0.0123 | -0.0583 |
| | RL | 0.4981** | 0.0137 | -0.0048 | -0.0991 |
| | LL | 0.4952** | 0.0302 | -0.0066 | -0.1012 |

**B. Boys**

| | | Sports test | | | |
|---|---|---|---|---|---|
| | | HG | 50S | SLJ | 20SR |
| Phase angle | RU | 0.0868 | -0.1402* | 0.1466* | 0.1228 |
| | LU | 0.1763** | -0.1389* | 0.0697 | 0.1155 |
| | RL | 0.2939** | -0.3120** | 0.3017** | 0.2797** |
| | LL | 0.3114** | -0.3498** | 0.3388** | 0.2994** |
| MW | trunk | 0.4001** | 0.00541 | 0.0807 | -0.0322 |
| | RU | 0.3279** | -0.0539 | 0.0291 | 0.1283 |
| | LU | 0.4033** | 0.0427 | 0.0991 | 0.0205 |
| | RL | 0.3768** | 0.0126 | 0.0992 | 0.0532 |
| | LL | 0.3910** | 0.0290 | 0.1101 | 0.0417 |

MW; muscle weight, RU; right upper limb, LU; left upper limb, RL; right lower limb, LL; left lower limb, HG; hand grip, 50S; 50m splint time, SLJ; standing long jump, 20SR; 20m shuttle run.

*: $p < 0.05$,

**: $p < 0.01$ (Pearson's product moment correlation coefficient).

Kwon et al. The Test of Gross Motor Development-Second Edition [15], the method used by Kwon et al. to assess motor skills, roughly assesses running and ball throwing and does not numerically record actual abilities such as running, jumping, throwing, and kicking. This divergence in assessment methods might account for the differing results between the two studies. Additionally, this study included a larger number of participants and a more uniform age range; therefore, it might have projected better real-world results than Kwon's report.

Several reports exist on the relationship between outdoor play and BF [16–18]; however, no previous reports have explored the relationship with MW or PhA. As mentioned in the introduction, PhA has been reported to be related to muscle quality [7–10], representing the geometric relationship between resistance and reactance and serving as raw data for BIA. In this study, PhA did not significantly differ according to the amount of time spent playing outside in girls. However, in boys, the PhA of the lower limbs was significantly greater in the group with more outdoor play. This suggests that in boys, more outdoor play improves the muscle quality of the lower limbs and not the upper limbs, and in girls, more outdoor play does not improve the muscle quality of the whole body. The causes of these sex differences are unclear; however, we speculate that sex differences in the content of outdoor play may be related. Girls who played outside had more MW, whereas no difference was observed between the two groups of boys. This trend contrasts with the pattern observed in PhA, and the reason

for this trend is unclear. However, it might be attributable to girls who play more outside having a greater HT and BW, as indicated in Table 1. Like PhA, future considerations should account for the content of outdoor play.

Various reports exist on physical performance and body composition, and the relationship between HG and muscle mass has long been pointed out [19–21]; however, correlations between other detailed physical performance and body composition are unknown. In this study, the PhA of the lower limb was significantly correlated with 50S, SLJ, and 20SR besides HG, with the correlation coefficients being greater in boys. HG was also significantly correlated with MW besides PhA. These results suggest that MW is related to muscle strength, similar to HG, whereas PhA correlates more strongly with movements that require complex coordinated movements, such as running and jumping. Previous studies have reported that PhA is associated with muscle quality [7–10], and the results of this study indicate that the ability to run or jump may be more related to muscle quality than to muscle strength alone. In a report on sex differences in the effects of outdoor play on physical activity, Kwon et al. reported that outdoor activity contributed more to physical activity in girls in their study of preschool children; however, the causes were not considered in detail [3]. Like sex differences and their effects on body composition on outdoor playtime, further investigation is needed regarding sex differences and the relationship between outdoor play, body composition, and physical function. This includes exploring external factors, like sex differences in the content of outdoor play, and internal factors, such as physiological differences between sexes.

This study had a few limitations. First, uniformity was lacking in the conditions during which body composition was measured. PhA is an evaluation index based on information obtained when a 50 kHz electric current is applied to the inside and outside of cells, and it varies with the amount of water in the body and the time elapsed after a meal. Participants in this study were not subjected to restrictions regarding fasting duration or water intake before the examination, and the potential impact of this factor on the results cannot be dismissed. Secondly, only children aged 8 years were examined. PhA increases and decreases with age in children [22,23]. Similar studies in different age groups may not yield significant results, and the results of this study may not be universally applicable to all age groups. We plan to conduct similar studies in different age groups to compare the results with those of the present study. Finally, complete data were available for only 494 out of the 1,340 participants in the cohort. The primary reason for the exclusion of so many participants was the inability to collect the school sports test results. The sports test results were collected by asking each participant to voluntarily submit the sports test results conducted at their elementary school. Therefore, it is believed that many participants forgot to record the results of the sports test or did not submit the results because it was inconvenient. However, the relatively low inclusion rate of participants had minimal impact on the overall results, given the absence of significant differences in body composition or time spent playing outside when comparing the groups with and without sports test results.

## Conclusions

Outdoor play habits improve muscle quality, particularly in boys, and this improvement may be related to enhanced athletic performance in children. The effect of outdoor play on girls differed from that of boys in that it affected muscle mass. Further research examining the sex differences in the results obtained in this study will clarify the relationship between physical activity and body composition, contributing to children's health.

*The conclusions of this article are solely the responsibility of the authors and do not represent the official views of the above government.*

## Supporting Information

**S1 Table. Minimal data set for the study.** This dataset includes anonymized data on the participants' body composition measurements, outdoor playtime, and physical performance test results.
(CSV)

## Acknowledgments

We are grateful to all the participants of the JECS-Y and all individuals involved in the data collection. We also thank the following members of the JECS-Y as of 2020: Zentaro Yamagata (Principal investigator, e-mail: zenymgt@yamanashi.ac.jp), Ryoji Shinohara, Sanae Otawa, Anna Kobayashi, Sayaka Horiuchi, and Megumi Kushima (Center for Birth Cohort Studies, Interdisciplinary Graduate School of medicine, University of Yamanashi, Chuo, Japan); Takeshi Inukai and Emi Sawanobori (Department of Pediatrics, School of Medicine, University of Yamanashi, Chuo, Japan); Kyoichiro Tsuchiya (Third Department of Internal Medicine, University of Yamanashi, Chuo, Japan); Takahiko Mitsui (Department of Urology, Interdisciplinary Graduate School of Medicine, University of Yamanashi, Chuo, Japan); Kenji Kashiwagi (Department of Ophthalmology, University of Yamanashi, Chuo, Japan); Daijyu Sakurai and Hiroyuki Watanabe (Department of Otorhinolaryngology-Head and Neck Surgery, School of Medicine, University of Yamanashi, Chuo, Japan); Koichiro Ueki and Naana Baba, (Department of Oral and Maxillofacial Surgery, Interdisciplinary Graduate School of Medicine, University of Yamanashi, Chuo, Japan); and Hiroshi Yokomichi, Kunio Miyake, Yuka Akiyama, Tadao Ooka, and Reiji Kojima (Department of Health Sciences, School of Medicine, University of Yamanashi, Chuo, Japan).

We would like to thank Editage (www.editage.jp) for English language editing.

## Author contributions

**Conceptualization:** Masanori Wako.

**Formal analysis:** Masanori Wako, Ryoji Shinohara.

**Investigation:** Masanori Wako, Taro Fujimaki.

**Project administration:** Ryoji Shinohara, Sanae Otawa, Sayaka Horiuchi, Megumi Kushima, Zentaro Yamagata.

**Supervision:** Zentaro Yamagata, Hirotaka Haro.

**Writing – original draft:** Masanori Wako.

**Writing – review & editing:** Taro Fujimaki, Jiro Ichikawa, Kensuke Koyama, Anna Kobayashi, Hirotaka Haro.

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
