## [Decision Letter · Decision Letter 0]

19 Nov 2024

PONE-D-24-13716Effects of outdoor play on body composition and physical performance in children: The Yamanashi Adjunct Study of the Japan Environment and Children’s StudyPLOS ONE

Dear Dr. Wako,

Thank you for submitting your manuscript to PLOS ONE. After careful consideration, we feel that it has merit but does not fully meet PLOS ONE’s publication criteria as it currently stands. Therefore, we invite you to submit a revised version of the manuscript that addresses the points raised during the review process.

We look forward to receiving your revised manuscript.

Kind regards,

Tadashi Ito

Academic Editor

PLOS ONE

**Journal Requirements:**

Reviewers' comments:

Reviewer's Responses to Questions

**Comments to the Author**

1. Is the manuscript technically sound, and do the data support the conclusions?

Reviewer #1: Yes

Reviewer #2: Yes

2. Has the statistical analysis been performed appropriately and rigorously? 

Reviewer #1: Yes

Reviewer #2: Yes

3. Have the authors made all data underlying the findings in their manuscript fully available?

Reviewer #1: Yes

Reviewer #2: Yes

4. Is the manuscript presented in an intelligible fashion and written in standard English?

Reviewer #1: Yes

Reviewer #2: Yes

5. Review Comments to the Author

**Reviewer #1:**  The manuscript entitled ‘Effects of outdoor play on body composition and physical performance in children: The Yamanashi Adjunct Study of the Japan Environment and Children’s Study’ assessed the data of 494 eight-year-old children in the Japan Environment and Children Study. The authors concluded that outdoor play significantly correlated with improved sports performance.

The manuscript is well-written. The reviewer has several comments on this manuscript.

Comments:

1. Why were the data of 8-year-old participants isolated and assessed? The authors should describe the reason in the Discussion section.

2. Why were 846 participants excluded from the 1,340 participants? The excluded number is too many to evaluate the overall trends of this study statistically. The exclusion criteria should be described in detail.

3. The definition of outdoor playtime should be described in detail. Does the outdoor playtime mean an average of a week or every day?

4. The reviewer recommends that the statistical analysis of correlations between sports tests and body composition should also be conducted by the total participants, not by boys and girls differentially.

**Reviewer #2:**  Correct the word "splinting" in page 8 Line6 to "sprinting"

Test of normality should be clarified in "Statistical analysis"

In "Discussion", some study limitations should be mentioned, such as determining the type of outdoor activities, whether these children live in urban or rural areas, did the child perform these outdoor activities daily, whether during school period or vacation and what is the maximal time for these outdoor activities.

6. PLOS authors have the option to publish the peer review history of their article (what does this mean? ). If published, this will include your full peer review and any attached files.

**Do you want your identity to be public for this peer review?** For information about this choice, including consent withdrawal, please see our Privacy Policy .

Reviewer #1: No

Reviewer #2: **Yes: ** Ahmed Mohamed Elnahhas

---

## [Author Response · Author response to Decision Letter 0]

9 Dec 2024

Reviewer 1

Comment１:

Why were the data of 8-year-old participants isolated and assessed? The authors should describe the reason in the Discussion section.

Response:

Thank you for your question. The Japan Environment and Children's Study is a nationwide birth cohort study on children's health and development that follows children from before birth to approximately 20 years of age and is conducted under the initiative of the Ministry of the Environment. Specifically, document surveys are conducted every 6 months after birth, and face-to-face surveys are conducted every few years. It was decided from the planning stage that a face-to-face survey with uniform content would be conducted nationwide at 8 years of age. In addition, our Koshin Unit Center independently conducted surveys on postural stability, dental and oral surgery surveys, and intestinal bacteria tests. This study used these data and targeted 8-year-old children. Some of these points have been added in the Materials and Methods section (lines 116–119).

Comment 2:

Why were 846 participants excluded from the 1,340 participants? The excluded number is too many to evaluate the overall trends of this study statistically. The exclusion criteria should be described in detail.

Response:

Thank you for pointing this out. We agree with you. The main reasons for the low collection rate of sports test results were that the timing of the survey coincided with the coronavirus disease epidemic and that many elementary schools did not have sports tests. The timing of the survey could not be changed because it was to be conducted at a predetermined period of the Ministry of the Environment’s cohort program. This information has been added to the paragraph on study limitations (lines 365–369).

Comment 3:

The definition of outdoor playtime should be described in detail. Does the outdoor playtime mean an average of a week or every day?

Response:

Thank you for your question. We used the results of a survey in which parents were asked if their children played outside for at least 1 hour per day (lines 152–153). Although this was a survey and may not be accurate in some respects, we believe that it is an acceptable form of questioning for making a determination of whether children play outside more or prefer to play indoors.

Comment 4:

The reviewer recommends that the statistical analysis of correlations between sports tests and body composition should also be conducted by the total participants, not by boys and girls differentially.

Response:

Thank you for your question. All the study results include not only boys and girls separately but also the total participants (Table 3a, lines 274–276).

Reviewer 2

Comment 1:

Correct the word "splinting" in page 8 Line6 to "sprinting"

Response:

Thank you for pointing this out. The spelling of “sprinting” has been corrected (line 145).

Comment 2:

Test of normality should be clarified in "Statistical analysis"

Response:

A description of normality has been added to the Materials and Methods section (lines 155–156). Since most data were normally distributed, t-tests and Pearson correlation coefficients were used in the statistical analyses.

Comment 3:

In "Discussion", some study limitations should be mentioned, such as determining the type of outdoor activities, whether these children live in urban or rural areas, did the child perform these outdoor activities daily, whether during school period or vacation and what is the maximal time for these outdoor activities.

Response:

Thank you for your comments. This study was based on data from a survey of parents who were asked “Do you play outside for at least 1 hour per day? The participants were recruited from all over our prefecture, including those living in mountainous areas and residential areas. As you pointed out, it is necessary to examine the content of outdoor play and the area (rural or urban) in which they live. However, in Yamanashi Prefecture, there are no extremely rural areas or extremely urban areas; thus, we do not believe that there are significant differences in outdoor play and lifestyles in the areas where the children live.

---

## [Decision Letter · Decision Letter 1]

8 Jan 2025

Effects of outdoor play on body composition and physical performance in children: The Yamanashi Adjunct Study of the Japan Environment and Children’s Study

PONE-D-24-13716R1

Dear Dr. Masanori Wako,

We’re pleased to inform you that your manuscript has been judged scientifically suitable for publication and will be formally accepted for publication once it meets all outstanding technical requirements.

Kind regards,

Tadashi Ito

Academic Editor

PLOS ONE

Reviewers' comments:

Reviewer's Responses to Questions

**Comments to the Author**

1. If the authors have adequately addressed your comments raised in a previous round of review and you feel that this manuscript is now acceptable for publication, you may indicate that here to bypass the “Comments to the Author” section, enter your conflict of interest statement in the “Confidential to Editor” section, and submit your "Accept" recommendation.

Reviewer #2: All comments have been addressed

2. Is the manuscript technically sound, and do the data support the conclusions?

Reviewer #2: Yes

3. Has the statistical analysis been performed appropriately and rigorously? 

Reviewer #2: Yes

4. Have the authors made all data underlying the findings in their manuscript fully available?

Reviewer #2: Yes

5. Is the manuscript presented in an intelligible fashion and written in standard English?

Reviewer #2: Yes

6. Review Comments to the Author

Reviewer #2: (No Response)

7. PLOS authors have the option to publish the peer review history of their article (what does this mean? ). If published, this will include your full peer review and any attached files.

**Do you want your identity to be public for this peer review?** For information about this choice, including consent withdrawal, please see our Privacy Policy .

Reviewer #2: **Yes: ** Ahmed Mohamed Elnahhas

---

## [Editor Report · Acceptance letter]

PONE-D-24-13716R1

PLOS ONE

Dear Dr. Wako,

I'm pleased to inform you that your manuscript has been deemed suitable for publication in PLOS ONE. Congratulations! Your manuscript is now being handed over to our production team.

Kind regards,

on behalf of

Dr. Tadashi Ito

Academic Editor

PLOS ONE